# Training in Hypoxia at Alternating High Altitudes Is a Factor Favoring the Increase in Sports Performance

**DOI:** 10.3390/healthcare10112296

**Published:** 2022-11-16

**Authors:** Ovidiu Dragos, Dan Iulian Alexe, Emil Vasile Ursu, Cristina Ioana Alexe, Nicoale Lucian Voinea, Petronela Lacramioara Haisan, Adelina Elena Panaet, Andreea Mihaela Albina, Dan Monea

**Affiliations:** 1Department of Physical Education, University of Alba Iulia, 510009 Alba Iulia, Romania; 2Department of Physical and Occupational Therapy, Faculty of Movement, Sports and Health, Sciences, “Vasile Alecsandri” University of Bacau, 600115 Bacau, Romania; 3Department of Physical Education and Sports Performance, Faculty of Movement, Sports and Health, Sciences, “Vasile Alecsandri” University of Bacau, 600115 Bacau, Romania; 4Doctoral School, National University of Physical Education and Sport Bucharest, 060057 Bucharest, Romania; 5Sport Science and Physical Educational Doctoral School, Social and Humanities Sciences University of Craiova, 200585 Craiova, Romania; 6Faculty of Physical Education and Sport, “BabeșBolyai” University, 400084 Cluj-Napoca, Romania

**Keywords:** altitude, hemoglobin, erythropoietin, hypoxia, maximum volume of oxygen

## Abstract

Training above 1800 m causes increases in hemoglobin, erythropoietin and VO2max values in the bodies of athletes. The purpose of this study is to prove that living at an altitude of 1850 m and training at 2200 m (LHTH+) is more effective than living and training at 2000 m (LHTH). Ten endurance athletes (age 21.2 ± 1.5 years, body mass 55.8 ± 4.3 kg, height 169 ± 6 cm, performance 3000 m 8:35 ± 0:30 min) performed three training sessions of 30 days, in three different situations: [1] living and training at 2000 m altitude (LHTH), [2] living at 1850 m and training at 2200 m (LHTH+), and [3] living and training at 300 m (LLTL). The differences in erythropoietin (EPO), hemoglobin (Hb) concentration, and VO2max values were compared before and at the end of each training session. Data analysis indicated that LHTH training caused an increase in EPO values (by 1.0 ± 0.8 mU/mL, p = 0.002 < 0.05.); Hb (by 1.1 ± 0.3 g/dL, p < 0.001); VO2max (by 0.9 ± 0.23 mL/kg/min, p < 0.001). LHTH+ training caused an increase in EPO values (by 1.9 ± 0.5 mU/ML, p < 0.001); Hb (by 1.4 ± 0.5 g/dL, p < 0.001); VO2max (by 1.7 ± 0.3 mL/kg/min, p < 0.001). At the LLTL training, EPO values do not have a significant increase (p = 0.678 > 0.050; 1 ± 0.1 mU/mL, 0.1 ± 0.9%.), Hb (0.1 ± 0.0 g/dL, 0.3 ± 0.3%), VO2max (0.1 ± 0.1, 0.2 ± 0.2%, p = 0.013 < 0.05). Living and training at altitudes of 2000 m (LHTH) and living at 1850 m training at 2200 m (LHTH+) resulted in significant improvements in EPO, Hb, and VO2max that exceeded the changes in these parameters, following traditional training at 300 m (LLTL). LHTH+ training has significantly greater changes than LHTH training, favorable to increasing sports performance. The results of this study can serve as guidelines for athletic trainers in their future work, in the complete structure of multi-year planning and programming, and thus improve the process of development and performance training.

## 1. Introduction

For maximum training efficiency, the best solutions are sought to produce physiological changes favorable to performance. One method studied in performance sports is altitude training [1]. A series of studies show that the amount of oxygen in the breathed air decreases along with the increase in altitude, and the human body adapts to the conditions of reduced oxygenation [2,3]. These changes take place at the hematological level and result in easier transport of oxygen [4,5]. The hematological change which leads to easier transport of oxygen is the increase in hemoglobin mass [1,4,5]. All hematological and non-hematological changes result in increasing sports performance [6,7,8]. Numerous studies show the beneficial effect of altitude training on sports performance [9,10,11]. Wilber, R.L.A. demonstrated that living high (2500 m) and training low (1250 m) for 28 days increases the hemoglobin mass by 5% and VO2max by 4% [12]. Rodriguez, F.A. et al. showed that classical training of 45 swimmers at an altitude of 2320 m led to an increase in hemoglobin mass by 7.2 ± 3% after a 4-week training course at altitude [13]. Specialists’ opinions are divided, with some studies stating that exposure to altitude does not bring major changes [14]. Gore, C.J. et al. highlighted the fact that athletes exposed to 4 weeks of intermittent exposure to hypobaric hypoxia (3 hours/day, 5 days/week at 4000–5500 m) had increased their erythrocyte volume index by 2.3% and hemoglobin mass by 1%, and the EPO concentration doubled in the first 6 hours. With all these changes, the conclusion was that intermittent exposure to hyperbaric hypoxia did not accelerate erythropoiesis despite the increase in serum EPO [14]. Some studies show physiological changes similar to those produced by altitude training [15,16,17]. Man, M.C. et al. present an alternative, although less explored, that has the potential to positively influence performance while avoiding some of the negative physiological consequences of hypoxia being sand training. Increases in VO2max and VMA after sand training were high (1.3 ± 0.1%; p < 0.001 and 1.2 ± 0.1%; p < 0.001), and the hemoglobin values also increased (3.3 ± 1.1%; p = 0.035) [15].

It is scientifically proven that physiological changes favorable to performance are triggered in the athletes’ bodies after 21 days [2,4,12]. As a result, it is very difficult to complete a training session in a hyperbaric chamber [18], and for this reason, specialists are looking for the best conditions for living and training at an altitude that produce favorable effects for increasing sports performance [11].

Therefore, were compared the differences in hemoglobin (Hb), erythropoietin (EPO) concentrations, and running capacity (VO2max) in highly trained runners after 30-day training cycles at 2000 m (LHTH), high-altitude living (1850 m), training at 2200 m (LHTH+) and traditional low-altitude training at 300 m (LLTL). It was hypothesized that training at LHTL and LHTH+ would significantly increase blood parameters and physical performance compared to training at LLTL and greater improvement will be seen after LHTH+ training over time.

## 2. Materials and Methods

### 2.1. Participants and Study Design

Participants were 10 male athletes (age 21.2 ± 1.5; 1 athlete 19 years and 7 months; 9 athletes 22 years ± 7 months, body mass 55.8 ± 4.3 kg, height 169 ± 6 cm) who specialized in half-distance and long-distance races, with 3000 m 8:35.00 ± 0:30. The athletes went through the same training program for 30 days, in three separate training cycles. The difference between the training cycles was six months. The main inclusion criteria were competition in a national level event and that the participants were free from musculoskeletal injuries for at least 6 months before each training cycle of the study.

In total, 15 athletes were selected for the study, but in the end, only 10 could participate, by observing the training and health protocol in all three training sessions. The athletes went through the same training program for 30 days across three training cycles, separated by one year.

### 2.2. Sample Dimension

The G*Power software V 3.0.10 (Dusseldorf, Germany) was used to determine the sample size (the minimum number of subjects).

Analysis Of Variance (ANOVA) with repeated measurements was chosen to determine the sample size (see the image above), with the following input parameters:Effect size f (ES = 0.58);Error probability α = 0.05;Power (1-β err prob) = 0.8;No. conditions = 3 (no. groups in the image);No. of repetitions = 2 (before and after);Correlation for repeated measurements = 0.5;Non-sphericity correction = 1.

The result indicates a minimum sample size of 9 subjects to ensure a test power of 82.46%.

Standard meals were offered to the athletes at each training camp. In addition, all the athletes adhered to the same regimen of vitamin and mineral supplements. Anthropometric measurements, blood draws, and running performance tests were taken the day before and on the first day after each 30-day training program to determine changes in hemoglobin concentration and running performance. Blood samples were collected after 12-h fasting at the same time of day (±1 h), at each moment. Samples were stored < 48 h at 2–8 °C before analyses.

### 2.3. Training Schedule

The athletes completed three training courses, as follows:

In the first stage, the athletes performed a training cycle for 30 days at Piatra Arsă, (G1) at an altitude of 2000 m (LHTH). In the second stage, the same group of athletes performed another training that took place in Font Romeu, where they lived at the National Altitude Training Centre (1850 m) and trained at an altitude of 2200 m (LHTH+). Finally, the third training cycle took place at an altitude of 300 m (LLTL).

An example of a training program is detailed in Table 1. The training program was repeated in all three training cycles.

Each training stage had 50 training sessions. A total of 520 ± 20 km was covered.

### 2.4. Maximum Speed and Aerobic Capacity

The five-minute test required the athletes to run at full capacity for five minutes. VMA = distance achieved (in km) × 12. The five-minute test was used to determine the VMA and the formula VMA × 3.5 was used to predict the VO2max because all subjects were over 18 years of age [19]. A previous study confirms that it is a reliable and practical indirect method of estimating individual aerobic fitness in a trained population [20].

Chamoux et al. 1996 [20,21,22] state that VMA determined on the field depends on the duration of the effort and therefore on the used protocol. Examining the relationship between running speed and the running time log set from the world race record pace shows a significant point at 4.97 min, suggested as the reference time for the VMA. By convention, VMA could be measured on the field by a 5-minute test, regardless of sport.

### 2.5. Hemoglobin Concentration

Hb was determined from venous blood collected before and after each 30-day training session using a Diagon D-Cell 60 automated hematologyanalyzer (Diagon Ltd., Budapest, Hungary).

### 2.6. Erythropoietin Concentration

Serum EPO concentration was determined by the commercially available Human EPO Quantikine™ IVD ELISA kit (R&D Systems, Minneapolis, MN, USA).

### 2.7. Statistical Analyses

Statistical analyses were performed using SPSS software (v.26, IBM, Armonk, NY, USA). ANOVA for repeated measurements was used to determine whether characteristics (for example, body weight, Hb, EPO, VO2max) remained similar before each 30-day training session. Changes in these characteristics within each training session were detected using one-way ANOVA. ANOVA repeated measurements were further used to determine differences in relative change (the percentage difference from pre-training) in characteristics across training camps. A Bonferroni correction was used for multiple comparisons on pairs, and the Greenhouse–Geisser correction was used when sphericity was violated. Normality was assessed using the Shapiro–Wilk test. Statistical significance was set a priori at p < 0.05.

### 2.8. Sample Size

Statistical analysis was performed using SPSS v.26 software. The one-way ANOVA test with repeated measurements was used to analyze the evolution of the values of the characteristics EPO, Hb, and VO2max, whose values were measured at the beginning and end of each 30-day training period. The athletes trained in training stages of 30 days at different altitudes: at 2000 m (LHTH)-G1, living at 1850 m and training at 2200 m (LHTH+)-G2, and 300 m (LLTL)-G3.

One-way ANOVA with repeated measurements was also used to determine the evolution of each characteristic mentioned above, as well as the differences between the average values of each characteristic, at the end of each training period, in which the athletes trained at altitudes LHTH, LHTH+, namely LLTL.

The Bonferroni correction was used to compare multiple pairs of values. Mauchly’s test was also used for sphericity, and the Greenhouse–Geisser test was used for correction when it was violated.

The data normality was checked using the Shapiro–Wilk test.

## 3. Results

### 3.1. Maximum Volume of Oxygen (VO2max-mL/kg/min)

LHTH resulted in a significant increase in the maximum volume of oxygen. It increased by 0.9 ± 0.23 mL/kg/min to 56.1 ± 2.2 mL/kg/min at the initial assessment and to 57.0 ± 2.2 g mL/kg/min at the final assessment, the percentage progress being 1.6 ± 0.5 mL/kg/min The increase is statistically significant, p < 0.001.

The maximal oxygen volume increased during LHTH+ by 1.7 ± 0.3 mL/kg/min, from 56.7 ± 2.0 mL/kg/min to 58.4 ± 2.1 mL/kg/min at the end of the training period. The percentage of increase is 2.9 ± 0.5%. The value p < 0.001 shows a statistically significant increase in the mean.

At LLTL, the average maximum volume of oxygen increased by 0.1 ± 0.1 mL/kg/min, from 56.9 ± 1.9 mL/kg/min to 57.0 ± 1.8 mL/kg/min after the end of the training period. In percentages, the increase is 0.2 ± 0.2%. The difference between the two means is statistically significant, p = 0.013 < 0.05.

The average values of the maximum oxygen volume recorded at the end of each training period at the three altitudes were compared with the one-way ANOVA test for repeated measurements. It turned out that there is at least one pair of average values with significant differences, p = 0.004.

There are statistically significant differences noticeable between the average values of the maximum oxygen volume between all pairs of values: LHTH vs. LHTH+, the difference of 1310 mL/kg/min, p = 0.047 < 0.05 and LHTH+ vs. LLTL, the difference of 1380 mL/kg/min, p = 0.001 < 0.05. Bonferroni adjustment was used for multiple comparisons.

It was determined that the *highest average increase in maximum oxygen volume at the end of the training period occurred in the cohort living at 1850 m and training at 2200 m*. The graph in Figure 1 shows the average relative (percentage) increases in the maximum oxygen volume for the preparation periods at the three altitudes.

### 3.2. Hemoglobin Concentration (Hb-g/dL)

Hb measured before the start of each training camp were similar (p = 0.145).

Each training session resulted in increases in hemoglobin values as follows.

LHTH increased by 1.1 ± 0.3 g/dL from 13.6 ± 0.4 mL/kg/min at the assessment before the training stage to 14.7 ± 0.5 g/dL at the assessment after the training stage, the percentage progress being 8.0 ± 1.8%. The increase is statistically significant, p < 0.001.

The average hemoglobin concentration LHTH+ increased by 1.4 ± 0.5 g/dL, from 13.6 ± 0.5 to 15.0 ± 0.7 g/dL at the end of the 30 days of training. The percentage increase is 10.9 ± 2.1%. The value p < 0.001 shows a statistically significant increase.

LLTH shows an increase in the average hemoglobin concentration of 0.1 ± 0.0 g/dL, from 13.7 ± 0.5 to 13.8 ± 0.5 in percentage, the increase being 0.3 ± 0.3%. The difference between the two means is statistically significant, p = 0.015 < 0.05.

The mean hemoglobin values recorded at the end of each training period at the three altitudes were compared with the one-way ANOVA test for repeated measurements. It turned out that there is at least one pair of mean values with significant differences, p ≤ 0.001.

Statistically significant differences were noticed between the average hemoglobin concentrations among all pairs of values: LHTH vs. LHTH+, a difference of 0.35 g/dL, p = 0.045 < 0.05, LHTH, vs. LLTL, a difference of 0.87 g/dL, p = 0.002 < 0.05, LHTH+, vs. LLTL, a difference of 1.22 g/dL, p < 0.001 < 0.05. Bonferroni adjustment was used for multiple comparisons.

It was also noted that the greatest increase in Hb concentration was achieved at the end of the training period at LHTH+. The graph in Figure 2 highlights the average relative (percentage) increases in hemoglobin concentration for the training periods at the three altitudes.

### 3.3. Erythropoietin Concentration

Differences in the erythropoietin (EPO) concentration before and after each training cycle were compared.

LHTH resulted in a significant increase in the average concentration of erythropoietin. This increased by 1.0 ± 0.8 mU/mL from 6.2 ± 1.9 at the initial assessment to 7.2 ± 2.5 at the final assessment, the percentage progress being 15.8 ± 8.4%. The increase is statistically significant, p = 0.002 < 0.05.

Following LHTH+, the mean erythropoietin concentration increased by 1.9 ± 0.5 mU/mL from 6.4 ± 1.8 to 8.3 ± 2.2 after 30 days of training. The percentage increase is 30.0 ± 5.6%. The value p < 0.001 shows a statistically significant increase.

LLTL does not bring a significant increase in the average concentration of erythropoietin, p = 0.678 > 0.05. In this situation, the average increase tends to be zero (0.1 ± 0.1 mU/mL), the average value measured before the beginning of the period is 8.9 ± 1.7 and the one at the end of the period is 9.0 ± 1.7. The percentage progress is 0.1 ± 0.9%.

Comparing the erythropoietin values recorded at the end of each training period, at the three altitudes, with the one-way ANOVA test for repeated measurements, it turned out that there is at least one pair of mean values with significant differences, p < 0.001.

There is only one statistically significant difference noticeable between mean erythropoietin concentrations in LHTH vs. LHTH+, p < 0.001, with Bonferroni adjustment for multiple comparisons.

It was also observed that *the highest increase in erythropoietin concentration was achieved at the end of the training period living at 1850 m and training at 2200 m*. This can also be seen from the graph in Figure 3, which shows the average percentage (relative) increases in erythropoietin concentration for the training periods at the three altitudes.

## 4. Discussion

This study aimed to assess changes in EPO, Hb concentrations, and running performance (VO2max) after 30 days, at a high altitude (2000 m), living at 1850 m and training at 2200 m, and the traditional training cycle at a 300 m altitude. Our main findings are that both high-altitude training and living at 1850 m and training at 2200 m resulted in significant improvements in EPO, Hb, and VO2max that exceeded changes in these parameters following traditional sea-level training. While training at high altitude caused greater relative increases in EPO and VO2max, living at 1850 m and training at 2200 m also resulted in greater increases in Hb, EPO, as well as sports performance (VO2max). The main findings of this study confirmed our initial hypothesis and suggest that the physiological changes triggered by 30 days of either high-altitude training or living at 1850 m and training at 2200 m are greater than those achieved by traditional training methods at the sea level.

The effect of training at high altitudes has been extensively researched [23,24,25,26,27,28,29]. Few studies have focused on high-altitude alternation [30]. Sharma, A.P. conducted research on running speed during training sessions wherein athletes lived at 2100 m and trained at 1400–2700 m, exercising over a range of intensities relevant to middle-distance running performance.

Movement speed in elite middle athletes is negatively affected at 2100 m natural altitude, with levels of impairment dependent on training intensity. Maintaining walking speed at certain intensities during altitude training may lead to greater perceived exertion [31].

The ten performance athletes in our study had a one-year training program individualized according to VMA values. Respecting the altitude adaptation period, at the end of the 30-day training period, the VO2max values increased by 1.6 ± 0.5% for LHTH and 2.9 ± 0.5% for LHTH+, which shows that the training burden incurred by a lack of oxygen leads to significant increases in sports performance.

Sports training at altitude can generally show an increase in aerobic capacity by 14% [32,33,34]. The acclimatization response to altitude depends on the individual physiological parameters, as is the case for the correlation between red blood cell mass that increases while the partial pressure of oxygen (PaO_2_) decreases [35]. This is why the “threshold altitude” for many people is 2200 to 2500 m [36]. For this reason, in this study was chosen the alternation of altitude; thus, living at 1850 m altitude, the altitude at which the hypoxia phenomenon appears [23], and training at 2200 m altitude, where the need for oxygen is bigger [36], were chosen for study. Acclimatization to environmental hypoxia generates cardiorespiratory adaptations reflected in the transport and use of oxygen [37], increases the concentration of hemoglobin in the blood, results in a high buffering capacity against the homeostasis of acid and basic elements of the body, has benefits for the structural and biochemical properties of skeletal muscle [37], and increases pulmonary ventilation. 

In our study, hemoglobin and erythropoietin concentrations at the end of the LHTH preparation period increased by 8.0 ± 1.8% and 15.8 ± 8.4% for LHTH+, the increases being higher: LHTH hemoglobin increases by 10.9 ± 2.1% and erythropoietin by 30.0 ± 5.6%, which supports the previous statements. On the other hand, maximal aerobic power is reduced as the variable changes; hematology compensates for this loss [38]. As mentioned above, altitude training may not be beneficial for everyone, as it initially depends on the handling and exposure time, which must be controlled according to the age, physical condition level, iron status, and energy requirements of each individual [33,39].

Stefan de Smet et al. [29] proved that gradual exposure to hypoxia in a normobaric chamber from 2000 m to 3250 m increased the total Hb mass by ~2.6% on average. Moreover, this increase was strongly correlated with the mean increase in sEPO, over the 5-week study intervention (r = 0.78, p < 0.05). In fact, the correlations between EPO and Hb mass gradually increased from the beginning (week 1, r = 0.54) to the end (week 5, r = 0.81) of the intervention period. This supports the rationality that maintaining high EPO levels during prolonged “living high” is crucial for increasing Hb mass at altitude. The current hypoxic protocol consistently increased sEPO concentrations from week 1 to week 5 in all subjects. This is likely due at least in part to the gradual increase in hypoxia from 16.4 to 14.0% FiO_2_ (~2000 to 3250 m), which resulted in even lower arterial oxygen saturations at week 5 than at week 1.

According to a recent meta-analysis, “living high” is expected to increase Hb mass by ∼1.08% for every 100 h of exposure to altitude above 2100 m [40]. This algorithm predicts a ∼4% increase in Hb mass over 350–400 h at >2100 m. Consequently, a hypoxic dose expressed as “hour-kilometers” predicts a ∼3.4% increase in Hb mass—linear model [41]. However, another study reported a 6.7% increase in Hb mass for only ~230 h of intermittent normobaric hypoxia, but at a constant altitude of 3000 m [42]. This could indicate that intermittent hypoxia may become more effective for increasing Hb mass at higher degrees of hypoxia. 

All these studies support the fact that living and training at altitude causes hematological changes that lead to increased sports performance.

Our study showed that there are different hematological changes influenced by the altitude at which the athlete lives and trains, changes that must be taken into account when planning training courses at altitude.

Our study is not without limitations. Although the study was conducted on elite runners, the sample size was small. Furthermore, it was measured the Hb concentration rather than Hb mass. Additionally, differences in weather and higher ambient temperatures at LHTH+ could have led to changes in Hb mass (with heat training).Another limitation would be the assumption that the physiological changes in the two parameters EPO and HB acquired in the first stage LHTH positively influence the changes produced in the second stage LHTH+, even if there is a period of six months between the two races, in which the athletes have trained at an altitude of 300 m. This remains to be studied in future research.

## 5. Conclusions

The results of this study can serve as guidelines for athletic trainers in their future work, in the complete structure of multi-year planning and programming, and thus improve the process of development and performance training.

Both alternatives of training at altitude are beneficial for increasing sports performance. Coaches and technical teams of athletes can introduce them to training plans according to geographical and financial possibilities.

The alternative of living at 1850 m and training at 2200 m is more expensive from a financial point of view as it involves daily ascent and descent to and from altitude; however, it produces greater changes and is therefore more beneficial for increasing sports performance.

In conclusion, this study shows that living at an altitude of 1850 m and training at 2200 m is more effective for performance athletes, due to several factors that remain to be studied, than living and training at an altitude of 2000 m.

## Figures and Tables

**Figure 1 healthcare-10-02296-f001:**
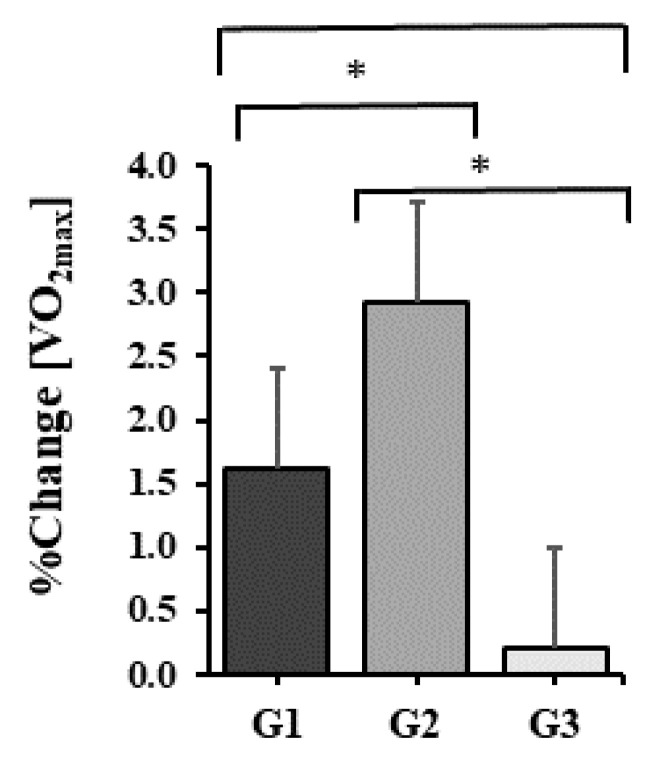
Relative before–after changes in VO2max concentration after 30 days of training at 2000 m altitude (G1), living at 1850 m and training at 2200 m (G2), namely 300 m (G3). Values displayed represent mean ± sd. Significant differences (p < 0.05) between training courses are marked with *.

**Figure 2 healthcare-10-02296-f002:**
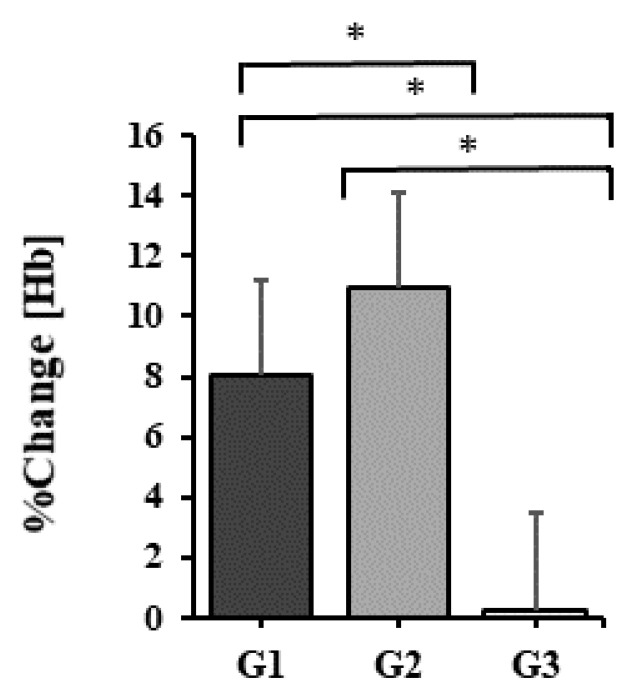
Relative before-after changes in Hb concentration after 30 days of training at 2000 m altitude (G1), living at 1850 m and training at 2200 m (G2), namely 300 m (G3). Values displayed represent mean ± sd. Significant differences (p < 0.05) between training courses are marked with *.

**Figure 3 healthcare-10-02296-f003:**
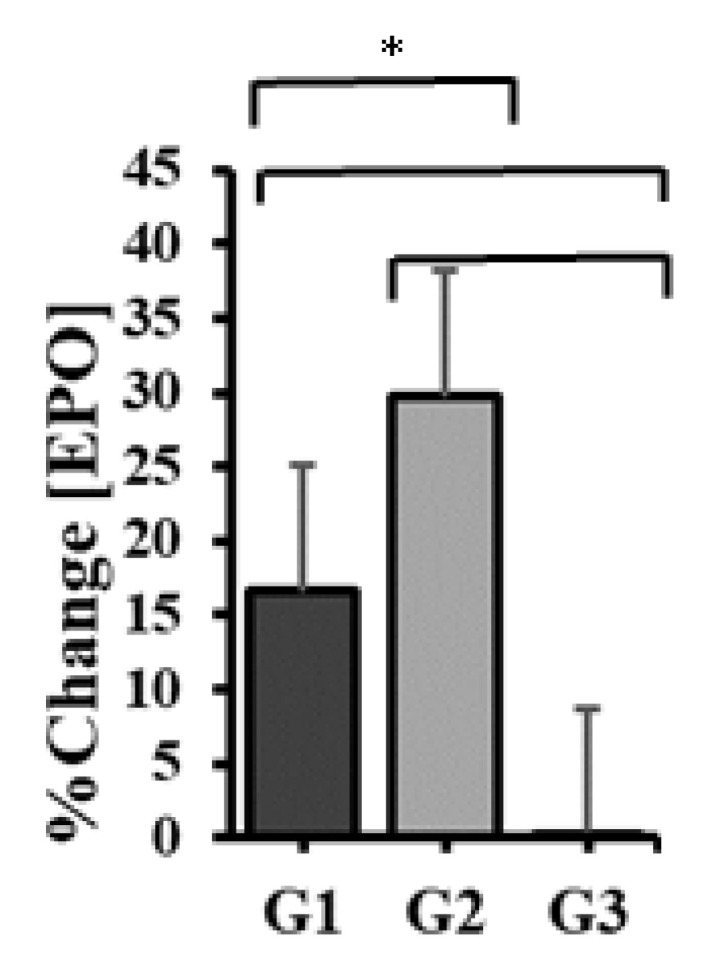
Relative before–after changes in EPO concentration, after 30 days of training at 2000 m altitude (G1), living at 1850 m and training at 2200 m (G2), namely 300 m (G3). Values displayed are mean ± sd. Significant differences (p < 0.05) between training courses are marked with *.

**Table 1 healthcare-10-02296-t001:** Example of a training program.

Day	Piatra Arsă-2000 m Altitude	Total Km-Running
1	S.T._1_ 4 km walking	4
2	S.T._2_ 6 km walking	6
3	S.T._3_ 6 km e.r.50% VMA/S.T._4_ 6 km e.r., segment strength-50% VMA	12
4	S.T._5_ 6 km e.r.50% VMA/S.T._6_ 8 km e.r., segment strength-60% VMA	14
5	S.T._7_ 8 km e.r.50% VMA/S.T._8_ 8 km e.r., segment strength-60% VMA	16
6	S.T._9_ 12 km r. uniform tempo, 60% VMA, segment. Strength	12
7	S.T._10_ 11 km e.r.50% VMA/S.T._11_ 6 km e.r. and 3 complete strength series-65% VMA	17
8	S.T._12_ 12 km r. uniform tempo, 65% VMA, segment. Strength/S.T._13_ 8 km r. uniform tempo, 10 × 100 m a.l.-65% VMA	20
9	S.T._14_ 14 km r. various land, segment strength and r.l.-70% VMA	14
10	S.T._15_ 14 km r. various land-75% VMA/S.T._16_ 8 km e.r., 3 series of ex. for strength	22
11	S.T._17_ 12 km r. uniform tempo, 70% VMA/S.T._18_ 10 km r. uniform tempo, 10 × 100 m r.l.-70% VMA	22
12	S.T._19_ 14 km r. uniform tempo, 75% VMA	14
13	S.T._20_ 12 km r. progressive various land 75–83% VMA/S.T._21_ 10 km r. uniform tempo, 70% VMA and 3 series of ex. for strength	22
14	S.T._22_ 6 km r. uniform tempo, 65% VMA, 30 × 100 m r. accelerated (100% VMA) with connection 100 m e.r. 4 km/S.T._23_ 10 km r. uniform tempo, stretching	24
15	S.T._24_ 8 km e.r., stretching 75% VMA/S.T._25_ 40 min r. (2min r. tempo sustained + 1 min conn. + 1 min r. tempo sustained + 1 min. connection) × 8 series (90% VMA)	24
16	S.T._26_ 14 km r. various land (75–80% VMA)	14
17	S.T._27_ 10 km r. various land and 10 × 100 m r.l. with 100 m e.r. 80% VMA/S.T._28_ 8 km r. uniform tempo, (75% VMA)	22
18	S.T._29_ 12 km r. tempo progressive-88–93% VMA/S.T._30_ 10 km e.r. (75% VMA)	22
19	S.T._31_ 14 km r. various land 80% VMA/S.T._32_ 10 km r. uniform tempo, (75% VMA)	24
20	S.T._33_ 12 km r. various tempo 92–94%VMA, 1 km e.r.	13
21	S.T._34_ 10 km e.r., 75% VMA/S.T._35_ 12 km e.r. segment strength (60% VMA)	22
22	S.T._36_ 16 km r. various land (65% VMA)	16
23	S.T._37_ 6 km e.r., 15 × 100 m with 100 m (95% VMA), 3 km e.r./S.T._38_ 10 km r. uniform tempo, 70% VMA	22
24	S.T._39_ 8 km e.r. 75%VMA/S.T._40_ 3 km e.r, 20 × 300 m with connection 100 m e.r. (30 s) 100%	20
25	S.T._41_ 2 h’ walk-forest	0
26	S.T._42_ 12 km r. 80% VMA/S.T._43_ 10 km r. uniform tempo, 75% VMA	22
27	S.T._44_ 12 km r. progressive various land 75–83% VMA/S.T._45_ 10 km r. uniform tempo, 70% VMA and 3 series of ex. for strength	22
28	S.T._46_ 6 km r. uniform tempo, 65% VMA, 30 × 100 m r. accelerated (100% VMA) with connection 100 m e.r. 4 km/S.T._47_ 10 km r. uniform tempo, stretching	24
29	S.T._48_ 8 km e.r., stretching 75% VMA/S.T._49_ 40 min r. (2min r. tempo sustained + 1 min conn. + 1 min r. tempo sustained + 1 min. connection) × 8 series (90% VMA)	24
30	S.T._50_ 16 km r. various land (65% VMA)	16

Abbreviations: r.—running; e.r.—easy running; r.l.—running launched; S.T. _1_—session of training no. 1; VMA—maximum aerobic velocity.

## Data Availability

Not applicable.

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
