# Peer review of "Training in Hypoxia at Alternating High Altitudes Is a Factor Favoring the Increase in Sports Performance"

_healthcare, 2022, doi:10.3390/healthcare10112296_

Round 1
Reviewer 1 Report
Authors congratulations on the article, is very interesting. I indicate some improvement suggestions that should be considered.
Introduction
Line 77: Remove the “.” after the “We hypothesized that”
Line 83: Remove “ after the end of the sentence: with 3000 m 8:35 .00 ±0:30
Discussion
Line 277: Remove the “.” after the “Our main findings are that”
Conclusion
Please add practical implications of the study and future studies.
Author Response
Dear Revierwer,
I sincerely thank you for your observations and advice.
I tried to answer as correctly and responsibly as I could
Point 1: Introduction
Line 77: Remove the “.” after the “We hypothesized that”
Line 83: Remove “ after the end of the sentence: with 3000 m 8:35 .00 ±0:30
Discussion
Line 277: Remove the “.” after the “Our main findings are that”
Response 1: Thank you for reporting the listed error. We have managed to make all the corrections.
Point 2: Conclusion
Please add practical implications of the study and future studies.
Response 2:
The purpose of practice in hypoxic conditions, is to stimulate the erythropoietin production which leads to an increase in hemoglobin values, that strengthens the immune system and stimulates the sympathetic. The altitude and the duration of the training period at high altitude, are an important part of the preconditions for achieving the best results in performance sports. Choosing to live at a low altitude (1850 m), but sufficient to produce the physiological changes indicated above and training at a high altitude (2200 m) produces physiological changes favorable to performance, greater than living and training at an altitude of 2000 m. All of these changes indicate that this research is in compliance with previous research and that remaining in hypoxic conditions, with that statistical significance, increases the number of erythrocytes (red blood cells) in the blood.
The results of this study, can serve as guidelines for athletic trainers in their future work in training the complete structure of a multi-year planning and programming, and thus to improve the process of development and performance training.
Both alternatives of training at altitude are beneficial for increasing sports performance, coaches and technical teams of athletes can introduce them into training plans according to geographical and financial possibilities.
The alternative of living at 1850 m and training at 2200 m is more expensive from a financial point of view, as it involves daily ascent and descent to and from altitude, but it produces greater changes, as a result it is more beneficial for increasing sports performance.
For the next studies, it remains to research the other factors that lead to these changes (sleep quality, oxygen saturation level) at the two altitudes of 1850 m and namely at 2000 m. The main factor determined by this study is hypoxia.
On behalf of the authors, I thank you for taking the time to help us and provide us with correction points.
Respectfully,
Dan Iulian Alexe

Reviewer 2 Report
Dear Authors
I have reviewed your paper with great interest.
I will accept your paper after a minimal revision.
My revision is:
Title: Very Good
Abstract: Very Good
Introduction and AIM: The problem and the aim are well descripting.
Results: Focus on and well described.
Discussion and Thread: effectiveness Focus ON.
The assessment of outcomes in the high altitudines that evaluate the metabolic aspect and oxadiative aspect as TNF alfa, cite and discuss this paper:
Ripani U, Bisaccia M, Meccariello L. Dexamethasone and Nutraceutical Therapy Can Reduce the Myalgia Due to COVID-19 - a Systemic Review of the Active Substances that Can Reduce the Expression of Interlukin-6. Med Arch. 2022;76(1):66-71. doi: 10.5455/medarh.2022.76.66-71.
References: Well chosen but to improve
Figures and Table: Very Good.
Author Response
Dear Reviewer,
Thank you for your appreciation.
Metabolic changes and the oxidative aspect are not the subject of our research, but we will certainly have them in view of future research regarding training at altitude.
We have also reviewed the references.
On behalf of the authors, I thank you for taking the time to help us and provide us with correction points.
Respectfully,
Dan Iulian Alexe

Reviewer 3 Report
L23-24 : maybe you can be less speculative
L25 : sessions or periods ?
L29 : period
L30-32 : please merge these sentences.
You have to present and to synthetize your main results and don’t list all results.
L39 : 1850 and 2200m ?
Based on your protocol and your results, I suggest to change the title which could be confusing. I mean you have to talk about differences on altitude level for living and training.
L57 : maybe an error of spelling
L64 : the sentence is too familiar
L67-71 : what it the link with sand training ? I would prefer to have some introduction on the LHTH+ topic.
L100-101 : so, I don’t understand if the training content/programm was the same or not for the 3 training periods ?
L136-167 : you have to include this in the statistics section. I think it is deserving a better presentation.
I think for each results, you don’t need to write 3 or 4 sentences. Just merge the sentences and precise if the result is significant or not.
Could you give other names for G1, G2 and G3 in order to know directly in which condition it is corresponding ?
The figures and tables are really informatives. So you don’t have to write a lot to present your results. Please synthetize and merge sentences.
L286 : you don’t need to repeat it again here.
L286-293 : maybe you can discuss more about it and make some comparison with the results of your study
In the next paragraphes, I don’t see a lot of clarity and especially I would wait some explanations of the possibles reasons for maximizing physiological adaptations with LHTH+. We could have this structure for the discussion section :
- Synthesis of main results
- Presentation of the other studies on altitude alternation
- Physiological adaptations when training is higher than living (or something similar)
- The impact of different hypoxia dose, etc… (or something similar)
- Other ?
L330-332 : maybe another sentence with some hypothesis or study suggestions ? Pratical applications for coaches ?
Author Response
Dear Reviewer,
We thank you for taking the time to study our research and for the observations made. In the future, we will promptly respond to your observations and suggestions
Point 1: L23-24 : maybe you can be less speculative.
Response 1: Thank you for the suggestion, I have modified the abstract.
Training above 1800 m causes increases in hemoglobin, erythropoietin and VO2max values in the body of athletes. The purpose of this study is to demonstrate that living at an altitude of 1850 m and training at 2200 m(LHTH+) is more effective than living and training at 2000 m (LHTH)
Point 2: L25 : sessions or periods ?
Response 2: periods of 30 days
Point 3: L30-32: please merge these sentences. L39 : 1850 and 2200m ?
You have to present and to synthetize your main results and don’t list all results.
Response 3:
Ten endurance athletes (age 21.2±1.5, body mass 55.8±4.3kg, height 169±cm, performance 3000m 8:35±0:30 min) performed three training periods of 30 days, in three different situations: 1. Lives and trains at 2000m altitude (LHTH), 2. Lives at 1850m, altitude trains at 2200m(LHTH+), 3. Living and training at 300m (LLTL). Were compared the differences in erythropoietin (EPO), hemoglobin (Hb) concentration and VO2max values before and at the end of each training session. Data analysis indicated that LHTH training caused an increase in EPO values (by 1.0±0.8mU/mL, p=0.002<0.05.); Hb (by 1.1±0.3g/dL),the increase is statistically significant, p<0.001, VO2max (by 0.9±0.2.3 mL/kg/min), the increase is statistically significant, p<0.001, LHTH+ training caused an increase in EPO values (by 1.9±0.5mU/Ml), the value p<0.001 shows a statistically significant increase; Hb (by 1.4±0.5g/dL), the p<0.001 value shows a statistically significant increase; VO2max (by 1.7±0.3 mL/kg/min) at the end of the training period, the p<0.001 value shows a statistically significant mean increase. At the LLTL training, EPO values do not have a significant increase (p=0.678>0.050. 1±0.1 mU/mL, 0.1±0.9%.) Hb (0.1±0.0 g/dL, 0.3±0.3%) VO2max (0.1±0.1, 0.2 ±0.2%, p=0.013<0.05).Living and training at altitudes of 2000m (LHTH) and living at 185m training at 2200m (LHTH+) resulted in significant improvements of EPO, Hb and VO2max that exceeded the changes in these parameters following traditional training at 300m (LLTL). LHTH+ training has significantly greater changes than LHTH training, favorable to increasing sports performance.
Point 4: L57 : maybe an error of spelling
Response 4: Rodriguez F.A. all, showed that classical training at an altitude of 2320 m, of 45 swimmers, led to an increase in hemoglobin mass by 7.2±3% after a 4-week training course at altitude [13].
Point 5: L67-71 : what it the link with sand training ? I would prefer to have some introduction on the LHTH+ topic.
Response 5: There are studies showing similar changes to LHTH+, one of which is sand training
Point 6: L100-101 : so, I don’t understand if the training content/programm was the same or not for the 3 training periods ?
Response 6: The training program was identical for all three training periods
Point 7: Could you give other names for G1, G2 and G3 in order to know directly in which condition it is corresponding? The figures and tables are really informative. So you don’t have to write a lot to present your results. Please synthetize and merge sentences
Response 7: I renamed the groups G1-LHTH, G2- LHTH+,G3-LLTL
Point 8: In the next paragraphs, I don’t see a lot of clarity and especially I would wait some explanations of the possible reasons for maximizing physiological adaptations with LHTH+. We could have this structure for the discussion section:
Response 8: Thank you very much for the suggestions. I have changed, based on your suggestions, the discussion part and I have introduced suggestions for coaches.
On behalf of the authors, I thank you for taking the time to help us and provide us with correction points.
Respectfully,
Dan Iulian Alexe

Reviewer 4 Report
The general idea of the manuscript is very intersting but the planning and design of the investigation has serious flaws
1) The authors did not indicate the time in between conditions. The training effects of the first condition might clearly influence the variables data of the second condition as changes in physiological parameters such as EPO or Hb take days/weeks to be noticed.
2) The presentation of the results is not clear
3) In the methods a much more detailed explanation of the procedures is needed (i.e. time between stages, abbreviations, training rutine......)
4) There are several English grammar mistakes that make the paper difficult to read and understand. It needs to be rewritten
Author Response
Dear Reviewer,
We thank you for taking the time to study our research and for the observations made. In the future, we will promptly respond to your observations and suggestions:
Point 1: The authors did not indicate the time in between conditions. The training effects of the first condition might clearly influence the variables data of the second condition as changes in physiological parameters such as EPO or Hb take days/weeks to be noticed.
Response 1: Thank you very much for your observation.
The research was carried out over a period of 1 year and 8 months. The break between 30-day training sessions is 6 months. The reason is that there are no changes caused by training sessions at altitude
Point 2: The presentation of the results is not clear
Response 2: I have modified this section
Point 3: In the methods a much more detailed explanation of the procedures is needed (i.e. time between stages, abbreviations, training rutine......)
Response 3: I have modified this section
Point 4: There are several English grammar mistakes that make the paper difficult to read and understand. It needs to be rewritten
Response4: I have corrected the paper regarding the English language
On behalf of the authors, I thank you for taking the time to help us and provide us with correction points.
Respectfully,
Dan Iulian Alexe

Reviewer 5 Report
Since the sample size is small, observations recorded for each subject at each of the training altitudes i.e. 2000 m, 2200 m and 300 m respectively ( longitudinal study) should have been tabulated and presented to bring clarity for better understanding. This can be done now.
Data on body mass of the athletes have been recorded and given in material & method section. Hope , such observations were also recorded along with other three parameters. If so, the same may be presented.
Though the nutrient intake has been constant at all the three training altitudes, however, the composition of diet, especially the calorific value and protein intake should be given.
Author Response
Dear Reviewer,
Thank you for appreciating our effort. The topic under discussion is one that always arouses interest. As addressed in many specialized studies, as interesting and always new, with the new discoveries from the multidisciplinary and interdisciplinary fields.
Metabolic consumption, and protein intake in particular, could be a new future approach to such studies.
Your idea has already piqued our interest for the near future.
On behalf of the authors, I thank you for taking the time to help us and provide us with correction points.
Respectfully,
Dan Iulian Alexe

Round 2
Reviewer 4 Report
The general idea of the manuscript is very intersting but the planning and design of the investigation has still some flaws
1) Add to the limiatations that the training effects of the previous condition might clearly influence the variables data of the second condition as changes in physiological parameters such as EPO or Hb take days/weeks to be noticed. So, results must be trated with caution
2) In the methods a much more detailed explanation is needed. (i.e. units are needed for age....line 90)
3) There are still vEnery important Eglish grammar mistakes that make the paper difficult to read and understand. It needs to be rewritten by a professional (native speaker) or a specialized company
4) References. Unify format (i.e. line 61)
Author Response
Response to Reviewer Comments
Point 1: Add to the limiatations that the training effects of the previous condition might clearly influence the variables data of the second condition as changes in physiological parameters such as EPO or Hb take days/weeks to be noticed. So, results must be trated with caution
Reponse 1: Thank you for the advice and clarifications. We have introduced limitations in the article regarding the influence of the first training stage at altitude on the physiological changes of EPO and HB in the second training stage.
”Another limitation would be the assumption that the physiological changes in the two parameters EPO and HB acquired during the first stage LHTH positively influence the changes produced in the second stage LHTH+, even if there is a period of six months between the two races, in which the athletes s -they trained at an altitude of 300m. This remains to be studied in future research””
Point 2: In the methods a much more detailed explanation is needed. (i.e. units are needed for age....line 90)
Reponse 2 : Thank you for the advice and clarifications.
“Participants were 10 male athletes (age 21.2 ± 1.5; 1 athlete 19 years and 7months; 9 athletes 22 years ± 7 months, body mass 55.8 ± 4.3 kg, height 169 ± 6 cm”
Point 3: There are still vEnery important Eglish grammar mistakes that make the paper difficult to read and understand. It needs to be rewritten by a professional (native speaker) or a specialized company
Reponse 3: Thank you for the recommendations. We turned to the specialized service of MDPI (English-Editing-Certificate-53519) and we hope that we have fixed those aspects that were not very clear from the beginning.
With your observations we become better.
Point 4: References. Unify format (i.e. line 61)
Reponse 4: Thank you for the advice and clarifications. I also standardized line 61 references

Round 3
Reviewer 4 Report
All aspects look better after the 2nd review.
The only minor correction I suggest is to avoid personal forms of the verb like "...we found...". Change it by passive sentences
Lines: 83, 86, 104, 203, 208, 234, 239, 263, 266, 304, 346
Author Response
Dear Reviewer,
Certainly, all the aspects suggested by you helped us a lot in our endeavor to propose to the readers of this journal a well-structured, easy to understand and interesting article.
For this aspect, once again you receive our thanks.
We changed the forms of the personal verbs with passive sentences
Best regards
Dan Iulian Alexe
